# Antibody Content against Epstein–Barr Virus in Blood Extracellular Vesicles Correlates with Disease Activity and Brain Volume in Patients with Relapsing–Remitting Multiple Sclerosis

**DOI:** 10.3390/ijms241814192

**Published:** 2023-09-16

**Authors:** Mireya Fernández-Fournier, MariPaz López-Molina, Gabriel Torres Iglesias, Lucía Botella, Beatriz Chamorro, Fernando Laso-García, Inmaculada Puertas, Antonio Tallón Barranco, Laura Otero-Ortega, Ana Frank-García, Exuperio Díez-Tejedor

**Affiliations:** Neuroimmunology and Multiple Sclerosis Unit, Department of Neurology, Neurological Sciences and Cerebrovascular Research Laboratory, Neurology and Cerebrovascular Disease Group, Neuroscience Area of Hospital La Paz Institute for Health Research—IdiPAZ (La Paz University Hospital—Universidad Autónoma de Madrid), 28046 Madrid, Spain; mplm1995@gmail.com (M.L.-M.); gabri_t13@hotmail.com (G.T.I.); luciabotella733@gmail.com (L.B.); beatriz.lapaz2@gmail.com (B.C.); fernilaso.9@gmail.com (F.L.-G.); inmapuertas@hotmail.com (I.P.); antonio.tallon@salud.madrid.org (A.T.B.); afrankg@gmail.com (A.F.-G.); exuperio.diez@salud.madrid.org (E.D.-T.)

**Keywords:** antibodies, biomarker, Epstein–Barr virus, extracellular vesicles, multiple sclerosis

## Abstract

We aimed to analyze whether EVs carry antibodies against EBV antigens and the possibility that they could serve as diagnostic and disease activity blood biomarkers in RRMS. This was a prospective and observational study including patients with RRMS with active and inactive disease and healthy controls. Blood EVs were isolated by precipitation. Titers of antibodies against nuclear (anti-EBNA1) and capsid (anti-VCA) EBV antigens in EVs and in plasma, as well as content of myelin antibodies in EVs were determined by ELISA. An exploratory analysis of correlations with clinical and radiological data was performed. Patients with RRMS had higher titers of anti-VCA inside EVs and free in plasma than healthy controls. Patients with active disease showed higher levels of anti-EBNA1 in EVs, but not in plasma, than patients with inactive disease. EV anti-VCA levels correlated with disease duration and with decreased brain volume structures—total brain, white matter, gray matter, cerebellum, hippocampus, —but not with T2/FLAIR lesion volume or EDSS, SDMT, or 9HPT. In addition, EV anti-VCA correlated with EV anti-MBP. The anti-VCA and anti-EBNA1 content in EVs could represent diagnostic and disease activity blood biomarkers, respectively, in RRMS.

## 1. Introduction

Multiple sclerosis (MS) is an inflammatory demyelinating disease of the central nervous system (CNS) with multifocal areas of demyelination [1]. There are two main disease forms, with the most common being a relapsing disease, in which periods of inflammatory activity alternate with periods of disease inactivity, named relapsing–remitting MS (RRMS) [2]. There is also a progressive form of the disease whereby disability accumulates independently of relapses, associated predominantly with neurodegeneration, named progressive MS (PMS) [2]. PMS is divided into primary PMS (PMS), which is infrequent, or secondary PMS (SPSM), which occurs as a late stage of RRMS occurring after years of disease inflammatory activity.

MS disease pathogenesis is complex, involving genetic susceptibility and environmental factors leading to the development of a pathologic autoimmune response against myelin, leading to myelin destruction, axonal loss, and focal inflammatory infiltrates [3]. MS appears to be the result of a misdirected immune response against one or several myelin proteins; however, despite decades of research, no clear antigenic target has yet been identified as the cause [4,5].

Strong epidemiological data, particularly the high infection prevalence in patients with MS, suggest that the Epstein–Barr virus (EBV) might be a prerequisite for MS development [6]. However, the underlying pathogenic mechanisms are still unclear [7]. EBV is a member of the human herpes virus family that is transmitted via saliva and infects pharyngeal epithelial cells and B cells in the underlying tissue. In B cells, EBV uncoats in the cytoplasm and transfers its DNA to the nucleus, where it can remain in a state of latency in which the viral genome can still be replicated along with cellular DNA. This state is called “deep” latency, given that the virus can be reactivated following B cell activation [8]. The mechanism by which EBV infection could trigger MS remains controversial [9]. EBV shares some structural properties with host proteins, so it might trigger self-perpetuating autoimmunity through molecular mimicry [10,11]. Alternatively, a lack of control of persistent EBV infection might favor the establishment of a dysregulated immune response [9,12]. In any case, B lymphocytes may generate antibodies against EBV that, through molecular mimicry, may also attack the myelin, and this could initiate an “active” phase of the disease.

Extracellular vesicles (EVs) are small bilayer membrane-wrapped particles that derive from the multivesicular body of their EV-originating cells and contain as cargo a wide variety of molecules. Recent research has focused on EVs as possible immune mediators, given that they can cross the blood–brain barrier due to their small size, between 30 and 200 nm, and carry a cargo implicated in disease pathogenesis [13]. Since EVs are readily available in the blood, they could serve as biomarkers in a wide variety of diseases [13,14] because they participate in the immune response and mimic the functions of their cells of origin. Due to their nature, EVs might be able to carry antibodies; however, this hypothesis has been scarcely explored. Recently, our group found that the EVs of MS patients distinctively carry antibodies against myelin [15]. Antibodies are produced and secreted by B cells, and B cells are known to secrete EVs; thus, EVs could be mediating antibody transfer and may be detectable in the blood, constituting a potentially novel blood biomarker in MS [15].

We aimed to investigate whether antibodies against EBV were present in EVs of patients with MS and their possible usefulness as blood biomarkers. We measured levels of EV antibodies against EBV in terms of activity and neurological status in patients with RRMS to shed light on their possible role in disease pathogenesis. We also cross-examined EV antibody load for autoantibodies against myelin.

## 2. Results

### 2.1. Patients and Controls: Clinical and Demographic Data

We included a total of 59 sequentially recruited patients with RRMS, of whom 35 had active disease, 24 did not have active disease, and 31 were healthy controls. Patients and controls were mostly women (74.4% active MS, 57.1% not active MS, 75% healthy controls) with no statistical differences between groups. There was no statistically significant difference in age between patients with no disease activity (43.4 ± 7.7 years) and patients with active disease (40.2 ± 8.4 years); however, healthy controls were younger than patients with MS (29.9 ± 8.7 years). Regarding disease duration, patients with inactive disease had RRMS for a significantly longer time period (18.4 ± 12.6 months vs. 4.8 ± 4.4 months, *p* = 0.01) than those with active disease. Regarding clinical parameters, there were no significant differences between patient groups (active vs. not active disease) regarding expanded disability status scale (EDSS) (1 (IQR 2) vs. 1.5 (IQR 2.5)), symbol digit modalities Test (SDMT) (49.5 ± 13.8 vs. 43.2 ± 20.8), or 9-hole peg test (9HPT) dominant and nondominant hand test results (21.5 ± 3.4 vs. 23.3 ± 2.1 and 25.6 ± 5.9 vs. 28.2 ± 10.8) (Table 1).

### 2.2. Extracellular Vesicle Characterization

The characteristic circular shape and double membrane of the EVs were observed by transmission electron microscopy (Figure 1A). EV-specific markers CD9, CD81, CD63, and ALIX were verified as present in the EVs isolated from plasma (Figure 1B),. The EVs showed a characteristic size of 30–200 nm, as seen by nanoparticle tracking analysis (NTA) (Figure 1C,D).

### 2.3. Higher EV Anti-VCA Titers Were Found in Patients Compared with Healthy Controls

We found that patients with RRMS had higher titers of anti-VCA inside both EVs and free in plasma than healthy controls (120.7 [Q1 = 84.0; Q3 = 166.0] UR/mL vs. 61.1 [Q1 = 39; Q3 = 129.1] UR/mL, *p* = 0.001; 123.2 [Q1 = 93.7; Q3 = 153] UR/mL vs. 74.8 [Q1 = 58.8; Q3 = 97.3] UR/mL, *p* ≤ 0.001). We found no differences in anti-EBNA1 titers in plasma or EVs between patients with MS and healthy controls (Figure 2A).

### 2.4. Higher EV Anti-EBNA1 Antibody Levels Were Found in Patients with Disease Activity

When comparing patients according to disease activity, we found that patients with active disease had higher titers of anti-EBNA1 in EVs (108.7 [Q1 = 60.9; Q3 = 144.5] UR/mL vs. 91.1 [Q1 = 51.1; Q3 = 129.5] UR/mL; *p* = 0.027), but not in plasma. We did not find significant differences in anti-VCA content in patients with active disease compared with patients without disease activity, albeit the former had a tendency toward higher anti-VCA levels. (Figure 2B).

### 2.5. Disease Status Did Not Modify EV Antibody Content

No correlations were found between either anti-VCA or anti-EBNA1 and patient clinical status (EDSS, 9HPT, or SDMT) (Figure 3).

### 2.6. EV Antibody against VCA Content Correlated with Disease Onset

Regarding time from disease onset, we found a positive correlation (R = 0.4, *p* = 0.02) between anti-VCA levels in EVs and time from disease onset (Figure 4A).

### 2.7. EV Anti-VCA Content Correlated with Brain Structure Volumes

Anti-VCA content in EVs but not in plasma correlated negatively with both global and regional brain volumes: total brain volume (R = −0.3, *p* = 0.03), white matter volume (R = −0.3, *p* = 0.04), gray matter volume (R = −0.3, *p* = 0.03), cerebellar volume (R = −0.3; *p* = 0.03), hippocampus (R = −0.3; *p* = 0.04), putamen (R = −0.3; *p* = 0.03), thalamus (R = −0.3, *p* = 0.04), and caudate nucleus (R = −0.3; *p* = 0.03) in T2 sequences (Figure 4B–I).

**Figure 4 ijms-24-14192-f004:**
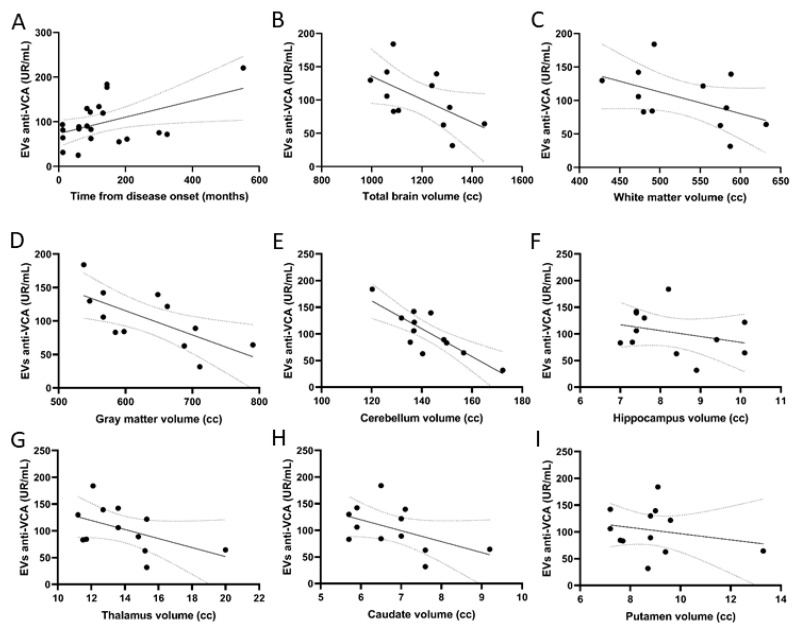
Relation of VCA antibody content in EVs and the time from diagnosis and brain structure volume. Correlation between anti-VCA levels in EVs and (**A**) disease duration, (**B**) total brain volume, (**C**) white matter volume, (**D**) gray matter volume, (**E**) volume of the cerebellum, (**F**) hippocampus volume, (**G**) volume of the thalamus, (**H**) caudate nucleus volume, and (**I**) volume of the putamen.

### 2.8. Antibodies against Myelin Basic Protein (MBP) in EVs of MS Patients Correlated with EV Anti-VCA Content

Anti-MBP content of EVs correlated with EV Anti-VCA content (R = 0.36; *p* = 0.006) (Figure 5A). No correlation was found between EV anti-MBP content and anti-VCA titers in plasma or with anti-EBNA1 titers in EVs or plasma. In addition, no correlation was found between anti-MOG content of EVs or plasma and antibodies against EBV or between EV content of a control protein (albumin) and EV antibodies against EBV (Figure 5B–J).

## 3. Discussion

This is a pilot exploratory study analyzing the content of antibodies against EBV in EVs in patients with MS and healthy controls. This is, to our knowledge, the first evaluation of antibody content against EBV in EVs, and the findings are of importance due to the growing, yet still unclear, possibility of EBV infection as an MS trigger.

MS is an inflammatory disease triggered by a poorly defined interplay of heritable and environmental factors, including EBV infection [9,16,17]. Our understanding of the mechanisms by which EBV infection leads to MS is still obscure and controversial [9,18]. Our data shed light on the possible mechanisms of EBV implication in MS pathogenesis, not only as a trigger for MS but also as a driver of disease activity throughout the disease course. This course, as our results indicate, could be mediated by EVs.

We found that patients with MS had significantly higher titers of anti-VCA antibodies in EVs and in plasma than healthy controls. Most studies have focused on anti-EBNA1 as a possible disease driver of MS [19], given that previous studies on reactivity against EBNA1 and VCA in patients found that plasma anti-EBNA1, but not anti-VCA, antibodies were associated with MS [19,20]. Therefore, few other studies have evaluated the response against EBV’s VCA as a biomarker in patients with MS. However, one previous study found both high-affinity anti-EBNA1 and anti-VCA IgG antibodies were more elevated in patients with RRMS and in patients with other inflammatory neurological disorders than in patients with noninflammatory neurological disorders [21]. This result indicates EBV could play a role in generating autoimmunity in neurological diseases. Another study found IgG titers in plasma to EBNA1 and VCA were increased in patients with MS with respect to healthy controls [22]. Along these lines, we found significant differences in the anti-VCA content of EVs in patients with MS compared with healthy controls. Our results suggest that antibodies against EVB found in EVs of patients could serve as a diagnostic biomarker in MS; however, this is a pilot study, and more studies should be carried out along these lines to analyze this.

Moreover, in the present study, we observed a correlation between anti-VCA and anti-MBP content in EVs but not with other myelin proteins (MOG) or control proteins (albumin). Given that EBV shares some structural properties with myelin proteins, this result may indicate that EVs could be loading both anti-EBV and anti-MBP antibodies to attack myelin during disease. On the other hand, recently, antibodies against myelin have been found in the EVs of MS patients at significantly higher levels than in healthy controls [15]. This argues in favor of the possible usefulness of the anti-VCA content of EVs as a blood biomarker in MS, suggesting that the EVs that load antibodies against EBV may also, through cross-reactivity, load autoantibodies against myelin antigens.

Furthermore, regarding the time from disease onset, we found a positive correlation between anti-VCA titers in EVs with time from disease onset, suggesting anti-VCA levels may increase as the disease evolves over time. Along these lines, we also found that anti-VCA titers in EV and in plasma correlated negatively with brain structure volumes. This finding is novel and warrants further investigation. Some studies have linked EBV infection with progressive forms of MS. EBV-infected B cell follicles and EBV gene expression have been reported in the brains of patients with MS, particularly in progressive forms of the disease [12,23,24,25]. Along these lines, we found increasing levels of antibody response to VCA in patients with increased disease duration and decreased brain volumes, both of which are surrogate markers for disease progression. We did not, however, find any correlation with EDSS, SDMT, or 9HPT.

We also found that patients with active disease had significantly higher anti-EBNA1 titers in EVs, but not in plasma, than patients with inactive disease, according to NEDA-3 criteria (no relapses, progression, or increased lesion load in the previous year). This finding suggests that there might be an association between disease activity and a reactivation of the antibody-dependent response against EBV. Studies have hypothesized that EBV could be latent in B lymphocytes, and viral reactivation might mediate antibody formation and, therefore, MS relapses [26]. A recent study evaluating total IgG antibodies purified from the brain of a patient with acute MS and CSF samples of patients with MS compared with inflammatory controls identified EBNA1 epitope reactivity to MS intrathecal antibodies corresponding to oligoclonal bands [27]. Another study evaluating CSF and OCB reactivity to EBNA1 had previously found patients with MS had significantly increased CSF/plasma antibody indices to EBNA1 [28]. The presence of OCB, indicating a B cell response with antibody production, is part of current MS diagnostic criteria [29], but the target antigen of these antibodies is not yet clear. Another study failed to identify differences in the oligoclonal EBV-specific antibody response between patients with MS and other inflammatory demyelinating diseases (mainly ADEM and NMO) and noninflammatory neurological controls [30]. That study did, however, conclude that the oligoclonal EBV-specific antibody response, when present, is mostly systemic in all neurological patients [30]. Our results could indeed indicate that EBV is involved in MS pathogenesis, and it might also drive the recurrent autoimmune response that occurs during MS relapses.

Furthermore, the fact that these titers were only elevated in EVs but not in plasma indicates that EVs might be mediating the reactivation of the autoimmune response. Cytokines and chemokines have long been known to be immune response mediators. Recent research has focused on EVs as novel particles mediating immune responses [31,32,33]. Our results indicate that the titers of anti-EBNA1 in the blood EVs of patients with MS could be involved in MS relapses and serve as a biomarker of disease activity.

The sample size of our study represents a limitation of our findings, and larger studies will be required to validate our conclusions. The fact that healthy controls were younger than either group of patients could be a confounding factor limiting the value of findings regarding levels of anti-VCA in EVs as diagnostic biomarkers in MS. However, titers of anti-VCA did increase with disease duration, which leads us to suspect that it does play a role in MS throughout the disease. Regarding healthy controls, it must also be noted that this study employed healthy blood donors as controls. It has been postulated that regular whole blood donation may have an impact on individual blood donors, particularly regarding cell counts and iron content [34]. However, recent mass-spectrometry-based studies have shown that only sparse changes occur in the plasma proteomes of new and regular donors after a whole blood donation [35]. In addition, healthy controls were younger than MS patients; however, they all had a positive serology to EVB. Also of note, patients with active disease had shorter disease duration than patients with inactive disease. This could be due to the fact that most patients have active disease initially when diagnosed, and it can take some time to find a treatment that keeps the disease stable, but some patients can remain stable with inactive disease for years. However, there were no significant differences in EDSS between groups. Of note, prior research indicates that EBNA1 reactivity might be particularly important in a subset of patients with MS. This finding, coupled with the fact that even if we did find significant differences in anti-EBNA1 content in EVs in MS patients according to disease activity, titers did vary, and ranges overlapped. These results could indicate that patient subsets should be further characterized, given that an anti-EBNA1 response could mediate disease activity only in a particular subset of patients, and this should be further explored in future studies with a greater sample size. However, we must also take into account that, in this study, patients were considered to have inactive disease when free of disease activity over a 1-year period. If patients with active disease had been compared with patients with no disease activity over a longer time frame, the differences found between both groups may have been greater.

In conclusion, we found that EVs carry antibodies against EBV antigens that could mediate disease processes, given that they are differentially found in patients with RRMS according to disease activity and with respect to healthy controls. Our results suggest that the EV content in anti-VCA and anti-EBNA1 antibodies could represent diagnostic and disease activity blood biomarkers, respectively, in patients with RRMS. Future research along these lines is necessary to validate our findings.

## 4. Materials and Methods

An observational and prospective pilot exploratory study was conducted at the Neuroimmunology and Multiple Sclerosis Unit of a tertiary university hospital in Spain between January 2020 and March 2023.

### 4.1. Patient Selection and Evaluation

Patients with RRMS were recruited sequentially, including both patients with active disease (defined by an ongoing relapse requiring steroid therapy) and patients without disease activity (NEDA 3 criteria: clinically and radiological stability for over a year). For comparison, we also included healthy controls. Patient inclusion criteria were diagnosis of RRMS, according to McDonald criteria [29], and age of ≥18 years. Patients were included before the start of corticosteroid treatment; steroid treatment in the prior 2 months was an exclusion criterion. Other exclusion criteria were progressive disease, pregnancy, lactation, drug or alcohol dependence, severe concomitant disease or another autoimmune disease, and participation in a clinical trial. All patients signed informed consent on inclusion.

Patients’ clinical status was evaluated using the EDSS as well as the 9HPT to assess hand motor and coordination function and the SDMT as a cognitive function screening tool.

To evaluate whether there was any clinical activity, as well as the neurological exam, patients underwent an MRI scan at a 1-year interval in search of increased T2 lesions or gadolinium-enhancing lesions. Anonymized images of these MRI scans, including a T1 3D sequence, were obtained and uploaded to aCloud. Using software FREESURFER 5.3.0, total and regional brain volumes, including T2-lesion load, white matter brain volume, gray matter volume, cerebellum, hypothalamus, and basal nuclei volumes were obtained whenever a 3D Flair sequence, without contrast, was available.

For comparison, healthy controls were recruited from blood donors attending the blood bank of the hospital and laboratory personnel after signing an informed consent document.

### 4.2. Sample Collection and Analysis

A total of 8 mL of peripheral blood was collected in 2 plasma-separator tubes from patients and controls. Samples were centrifuged at 3000 g for 15 min at 4 °C. The plasma obtained from each patient was aliquoted in Eppendorf tubes and stored at −80 °C until analysis.

### 4.3. Extracellular Vesicle Isolation

EVs were extracted from 50 µL of plasma using ExoQuick EV precipitation solution (System Biosciences, Palo Alto, CA, USA), as previously described [36], employing an additional purification step. Following the manufacturer’s instructions, 13 μL of ExoQuick buffer (System Biosciences) was added to 50 μL of plasma and incubated at 4 °C for 30 min. It was then centrifuged at 1500× *g* for 30 min to sediment EVs. Lastly, the supernatant was removed, and the pellet was resuspended in 100 μL of sterile PBS.

### 4.4. Extracellular Vesicle Characterization

NTA employs light scattering and Brownian motion properties to analyze EV size and quantity in a liquid suspension, as previously described [14]. In the present study, NTA using the NanoSight LM10 system (Malvern Panalytical, Malvern, UK) was performed. We analyzed EVs diluted in 300 µL of phosphate buffered saline (PBS) in a working concentration of 1 × 107 × 109 particles/mL and recorded three 60 s videos at a shutter speed of 30.00 ms and camera level of 13. For the analysis, we employed a detection threshold of 3 in order to analyze 50 nm vesicles or larger, and each sample was run three times.

Transmission electron microscopy was used to visualize the EVs and verify that their size ranged from 30 to 150 nm, as previously described [14]. The EVs were fixed in 2.5% glutaraldehyde 0.1 M sodium cacodylate solution for 1 h at 4 °C and postfixed with 2% osmium tetroxide for 1 h at 4 °C. The EV pellet was dehydrated with a graded acetone series and then embedded in resin. Thereafter, 60 nM thick sections were cut and observed under transmission electron microscopy at 80 kV, and the process was run in triplicate.

Western blot analysis was used to determine presence or absence of EV proteins (ALIX, CD63, CD9, and CD81), as previously described [14]. EVs were lysed with a radioimmunoprecipitation assay buffer (RIPA) (89900, Thermo Scientific, Waltham, MA, USA) and subjected to Western blot analysis using a 4–10% sodium dodecyl sulfate (SDS)-polyacrylamide gel (PAGE) for electrophoresis with 20 µg of protein per lane (quantified by bovine serum albumin). The following antibodies were used: anti-CD9 (Abcam, Cambridge, UK), antiCD81 (Abcam, Cambridge, UK), anti-CD63 (Abcam, Cambridge, UK), and anti-Alix (Cell Signal, Danvers, MA, USA). This procedure was run in triplicate.

### 4.5. Extracellular Vesicle Fragmentation for Analysis of Content in Antibodies

The EV membrane was disrupted using a pierce radioimmunoprecipitation assay buffer (RIPA) (Thermo Scientific, USA) 1X dilution and 20 min incubation. To promote EV membrane rupture, vortexing was carried out every 5 min.

### 4.6. Determination of EV Content in Antibodies against EVB

Titers of antibodies against both nuclear (anti-EBNA1-IgG) and capsid (anti-VCA-IgG) EBV antigens, both in blood-isolated EVs and in plasma, were determined by sandwich enzyme-linked immunosorbent assay (ELISA) using Euroimmun commercial kits via semiquantitative analysis (following manufacturer’s instructions) in all patients and healthy controls.

### 4.7. Determination of EV Content in Autoantibodies against Myelin Antigens

Anti-MBP, anti-MOG antibodies, and anti-albumin (employed as a negative control) were detected and quantified by ELISA (Abbexa, Cambridge, UK).

### 4.8. Data Analysis

Data were analyzed using the statistical IBM SPSS 23 program and were tested for normality using the Kolmogorov–Smirnov test for groups with more than 30 degrees of freedom and the Shapiro–Wilk test for groups with fewer than 30 degrees of freedom. Demographics, clinical data, and antibody titers were compared between groups using Student’s *t*-test for independent samples. The results are expressed as mean ± standard deviation (SD). An exploratory analysis of correlation of antibody titers with clinical and radiological data was performed using the Pearson correlation test for parametric data and Spearman correlation test for non-parametric data. *p*-values < 0.05 were considered significant at a 95% confidence interval.

## Figures and Tables

**Figure 1 ijms-24-14192-f001:**
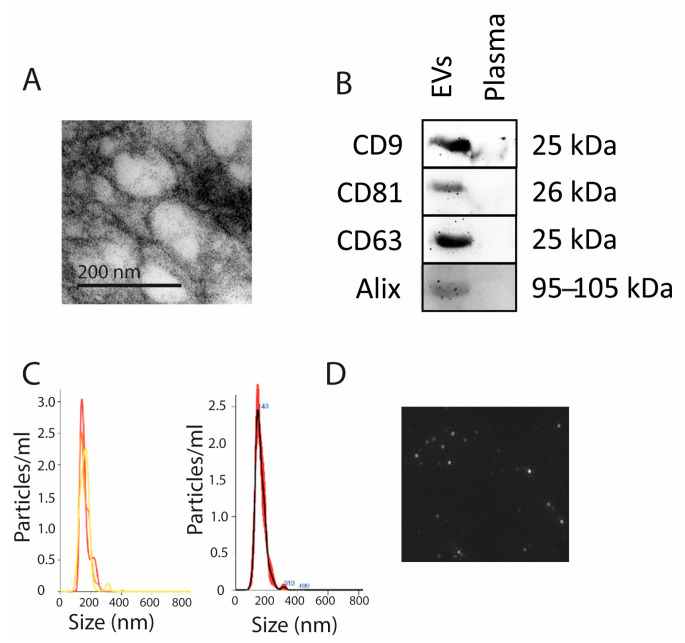
EV characterization. (**A**) Electron microscopy showing EVs of <200 nm. (**B**) CD9, CD81, CD63, and Alix markers present in EVs. (**C**) Graph showing EV sizes ranging 100–200 nm. The sample was loaded three times, shown in left panel (red, yellow, and orange lines). The mean (black line) and SD (red line) are shown in the right panel. (**D**) EVs as seen by NTA.

**Figure 2 ijms-24-14192-f002:**
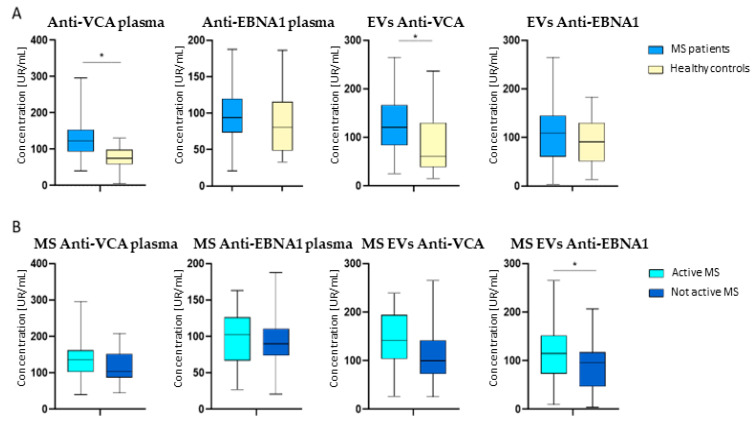
EVB antibody content in MS patients. (**A**) Comparison of EVB titers for VCA and EBNA1 in plasma and EVs between MS patients and healthy controls. (**B**) EVB antibodies for VCA and EBNA1 in plasma and EVs between MS patients according to disease activity. * = *p* ≤ 0.05.

**Figure 3 ijms-24-14192-f003:**
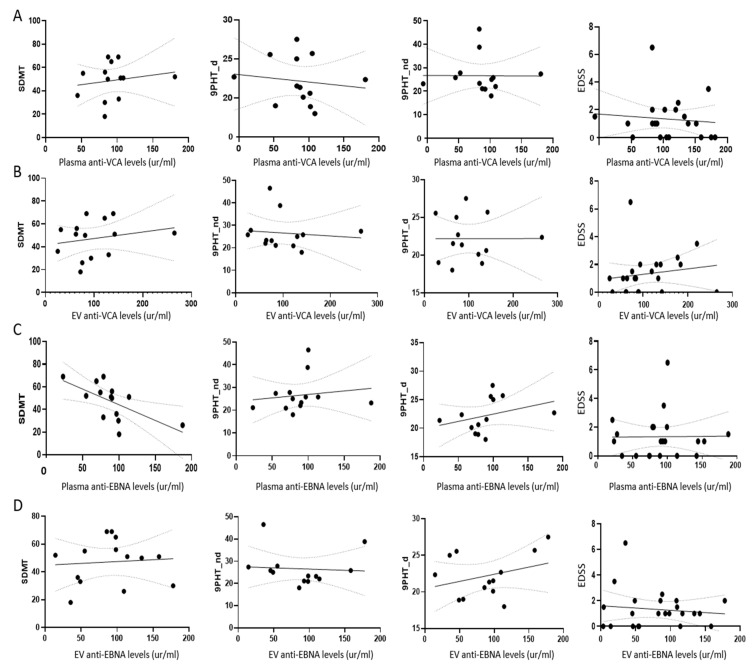
Relation between EBV antibody levels and disease status. (**A**) Plasma anti-VCA antibody levels. (**B**) EVs anti-VCA content. (**C**) Plasma anti-EBNA levels. (**D**) EVs anti-EBNA content.

**Figure 5 ijms-24-14192-f005:**
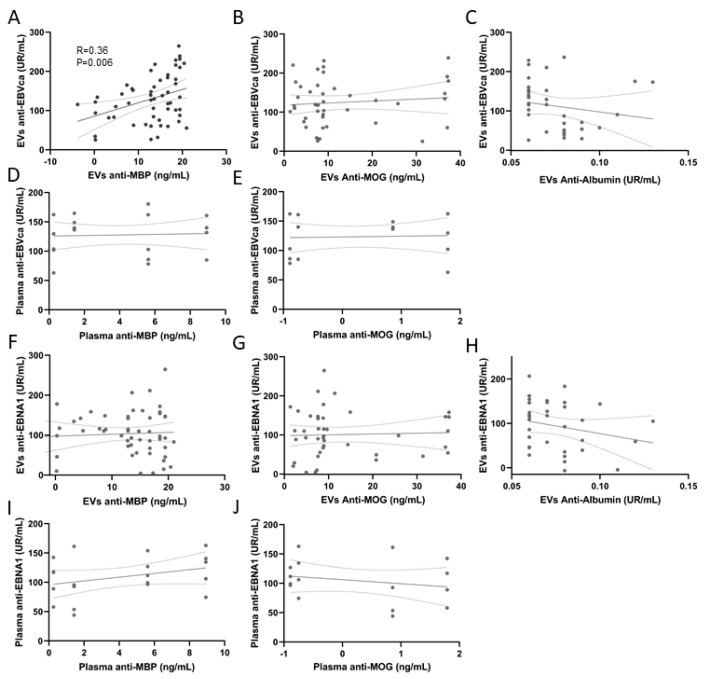
Relationship between EBV antibody content and anti-myelin antibodies in EVs and plasma. Correlation between anti-VCA levels in EVs and (**A**) anti-MBP, (**B**) anti-MOG, and (**C**) albumin in EVs. (**D**) Correlation between anti-VCA antibodies in plasma with anti-MBP and (**E**) anti-MOG in plasma. Correlation between anti-EBNA1 levels in EVs and (**F**) anti-MBP, (**G**) anti-MOG, and (**H**) albumin in EVs. (**I**) Relationship between anti-EBNA1 antibodies in plasma with anti-MBP and (**J**) anti-MOG in plasma.

**Table 1 ijms-24-14192-t001:** Clinical and demographic data.

	Active MS	Not Active MS	Healthy Control	
Sex (women, %)	74.4	57.1	75.0	*p* > 0.05
Age, years (mean ± SD)	40.2 ± 8.4	43.4 ± 7.7	29.9 ± 8.7 *	* *p* = 0.001
Time from disease onset (mean ± SD)	4.8 ± 4.4	18.4 ± 12.6		* *p* = 0.01
EDSS	1 (IQR 2)	1.5 (IQR 2.5)		*p* > 0.05
SDMT	49.5 ± 13.8	43.2 ± 20.8		*p* > 0.05
9HPT_DH (s)	21.5 ± 3.4	23.3 ± 2.1		*p* > 0.05
9HPT_NDH (s)	25.6 ± 5.9	28.2 ± 10.8		*p* > 0.05

SD, standard deviation; EDSS, Expanded Disability Status Scale; SDMT, Symbol Digit Modalities Test; 9HPT, Nine-Hole Peg Test; DH, dominant hand; NDH, nondominant hand; s, seconds.* = *p* ≤ 0.05.

## Data Availability

Data are available upon reasonable request.

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
