# Peer review of "Antibody Content against Epstein–Barr Virus in Blood Extracellular Vesicles Correlates with Disease Activity and Brain Volume in Patients with Relapsing–Remitting Multiple Sclerosis"

_ijms, 2023, doi:10.3390/ijms241814192_

Round 1
Reviewer 1 Report
The authors are researching one of the most important factors, EBV, to emerge recently as a risk factor in multiple sclerosis that has gathered a lot of supporting evidence from many groups around the world. Extracellular vesicles represent a novel advance in the understanding of how antibodies might be a bio-marker or a causative factor in autoimmunity. It is a pilot study with some promising observations linking EV antibodies to elements of MS disease. Several major and minor issues were identified in the manuscript.
major issues
Manuscript should clarify the definition of active and stable forms of RRMS, and how does this relate to relapsing versus remitting phase of disease. National MS Society suggests active, not active, worsening or not worsening as terminology for RRMS. Using a period of 1 year to determine remission is less than some other publications that use at least 2 years of data to determine if disease is in remission. Later in the paper, stable RRMS is equated to disease progression, and the stable MS group have 13 years more disease durations, although the EDSS is the same between groups. The healthy control are younger, which makes it difficult to assess the data since the main findings are between HC and MS. If patients had steroids in last 2 months should be excluded.
Oligoclonal bands intrathecal (lines 58 - 65), are generally considered to be non-pathogenic, stronger evidence from literature should be provided to support the claim that they are pathogenic. Since the study looks at blood immunoglobulins, not intrathecal, is a good approach for biomarkers, rather than a way to assess pathgenicity of intrathecal oligoclonal bands. Suggest revising the introduction to focus on blood biomarkers.
minor issues
typos Sclerosis should not be capitalized (line 31). superscripts missing line 274.
What is a detection threshhold of 3 mean ( line 276)
This procedure was run in triplicate line 290 - clarify, were the same sample run on three lanes, or were three different samples analyzed.
Methods 4.5 not much detail, seems like the same buffer as previous section.
The healthy controls provided informed, signed consent (line 250). It was not mentioned in methods that the patients provided informed, signed consent, although the statement at the end says everyone did. Should specifiy in methods.
Figure 1 A There is a white line cutting through image. B A loading control blot or stain should be shown to verify the plasma had sample on the membrane. More of the blot should be shown, in the figure or in supplemental to assess the possible crossreactivity of antibody, and to see the molecular weight markers. C) not clear why two graphs are shown, what are the colours (red, yellow, black, grey), D) missing scale bar.
Discussion, line 175 claims that the results indicate EV antibodies can be diagnostic biomarker of MS, however, it is powered to be a pilot study, and the data shows considerable overlap between the MS and HC despite being significantly different, and there were few differences between active or stable.
Writing was very good. Just typo Figure 3 caption, VEB... perhaps meant to say EBV antibodies. Typo on x axis volumen
Author Response
Reviewer 1
The authors are researching one of the most important factors, EBV, to emerge recently as a risk factor in multiple sclerosis that has gathered a lot of supporting evidence from many groups around the world. Extracellular vesicles represent a novel advance in the understanding of how antibodies might be a bio-marker or a causative factor in autoimmunity. It is a pilot study with some promising observations linking EV antibodies to elements of MS disease. Several major and minor issues were identified in the manuscript.
The authors want to thank reviewer 1 for pointing out the interest of our work, in the light of rising evidence of EBV as a pivotal risk factor for MS. Also, we want to thank reviewer 1 for his/her time reviewing the manuscript and all comments and suggestions that have greatly helped ameliorate the accuracy of our work. A point-by-point response is provided below.
Major issues
Manuscript should clarify the definition of active and stable forms of RRMS, and how does this relate to relapsing versus remitting phase of disease. National MS Society suggests active, not active, worsening or not worsening as terminology for RRMS. Using a period of 1 year to determine remission is less than some other publications that use at least 2 years of data to determine if disease is in remission. Later in the paper, stable RRMS is equated to disease progression, and the stable MS group have 13 years more disease durations, although the EDSS is the same between groups. The healthy control are younger, which makes it difficult to assess the data since the main findings are between HC and MS. If patients had steroids in last 2 months should be excluded.
Thank you for pointing out that there are some issues that may eventually lead to confusion. Indeed, the National MS Society, in line with the wording of Lublin et al. in “Defining the clinical course of Multiple Sclerosis”1, suggests active vs. not active as wording to describe activity. We have thus adopted this terminology and the word “stable” has been replaced by either “not active” or “inactive”, alternatively, throughout the manuscript for clarity.
With respect to the time frame, current guidelines1,2 suggest disease activity should be assessed annually both clinically and by MRI. Since we are studying possible markers of disease activity, we have followed these guideline analyzing the data at a one-year interval.
- (manuscript ref. nº2 - introduction): Lublin FD, Reingold SC, Cohen JA et al. Defining the clinical course of multiple sclerosis: the 2013 revisions. Neurology. 2014 Jul 15;83(3):278-86.
- (manuscript ref. nº29 - methods): Thompson AJ, Banwell BL, Barkhof F et al. Diagnosis of multiple sclerosis: 2017 revisions of the McDonald criteria. Lancet Neurol. 2018 Feb;17(2):162-173.
We again thank the reviewer for pointing out the use of wording that can be confussing. MS is a disease that changes over time, with RRMS evolving to SPMS over the years/decades, in up to 80% patients. We have now explained in the introduction section that SPSM occurs as a late stage of RRMS for the benefit of readers who are not so familiar with this disease. Also, we have changed the fourth paragraph of the Discussion section as follows:
- We have changed “…suggesting anti-VCA levels may increase as the disease progresses” to “suggesting anti-VCA levels may increase as the disease evolves over time” (line 258)
- We have removed the sentence “Our results suggest that the anti-VCA content of EVs in patients might not only serve as a diagnostic biomarker, but it could also be a marker of disease progression” as we agree that this has not been studied. Specially so since, as the reviewer justly points out, no differences in EDSS were found between groups, and also no correlations were found with EDSS, SDMT or 9PHT scales.
Thus, this paragraph now reads as follows (lines 257-267):
“Furthermore, regarding the time from disease onset, we found a positive correlation between anti-VCA titers in EVs with time from disease onset, suggesting anti-VCA levels may increase as the disease evolves over time. Along these lines, we also found that anti-VCA titers in EV and in plasma correlated negatively with brain structure volumes. This finding is novel and warrants further investigation. Some studies have linked EBV infection with progressive forms of MS. EBV-infected B-cell follicles and EBV gene expression have been reported in the brains of patients with MS, particularly in progressive forms of the disease [12,23–25]. Along these lines, we found increasing levels of antibody response to VCA in patients with increased disease duration and decreased brain volumes, both of which are surrogate markers for disease progression. We did not, however, find any correlation with EDSS, SDMT or 9PHT.”
We would like to clarify that the difference in disease duration between patients with active disease and those with inactive disease is to be expected taking into account that patients are diagnosed with RRMS after a relapse, and that it can take some time to find a treatment that works and keeps them stable. However, the fact that the EDSS was similar between groups makes them comparable. We have included the following comment in this regard in the limitations section (lines 326-331):
“Also of note, patients with active disease had a shorter disease duration than patients with inactive disease. This can be due to the fact that most patients have active disease initially when diagnosed and it can take some time to find a treatment that keeps the disease stable, but some patients can remain stable with inactive disease for years. However, there were no significant differences in EDSS between groups.”
It is true that healthy controls are younger and this has been highlighted when analyzing the limitations of this pilot study. However, all controls had a positive serology for EBV, and this makes them good controls. We have now included a comment on this age difference in the limitations section (lines 325-326).
“Also, healthy controls were younger than MS patients, however they all had a positive serology to EVB.”
Regarding exclusion criteria based on prior steroid use, this has now been specified. All patients with active disease included in the study were relucted, and had their blood drawn, before steroid therapy was administered. No patients had received steroid therapy in the previous 2 months. A paragraph specifying inclusion and exclusion criteria has now been included at the top of section 4.1 (lines 335-364) as follows:
“Patients with RRMS were recruited sequentially, including both patients with active disease (defined by an ongoing relapse requiring steroid therapy) and patients without disease activity (NEDA 3 criteria: clinically and radiological stability for over a year). For comparison, we also included healthy controls. Patient inclusion criteria were: diagnosis of RRMS, according to McDonald criteria [29] and age of ≥18 years. Patients were included before the start of corticosteroid treatment, steroid treatment in the prior 2 months was an exclusion criteria. Other exclusion criteria were progressive disease, pregnancy, lactation, drug or alcohol dependence, severe concomitant disease or another autoimmune disease, and participation in a clinical trial. All patients signed informed consent on inclusion.”
Oligoclonal bands intrathecal (lines 58 - 65), are generally considered to be non-pathogenic, stronger evidence from literature should be provided to support the claim that they are pathogenic. Since the study looks at blood immunoglobulins, not intrathecal, is a good approach for biomarkers, rather than a way to assess pathgenicity of intrathecal oligoclonal bands. Suggest revising the introduction to focus on blood biomarkers.
We thank the reviewer for pointing this out. The athors have now rewritten the introduction section accordingly. It now reads as follows:
“Multiple sclerosis (MS) is an inflammatory demyelinating disease of the central nervous system (CNS) with multifocal areas of demyelination [1]. There are two main different disease forms, with the most common being a relapsing disease where periods of inflammatory activity alternate with periods of disease inactivity, named relapsing-remitting MS (RRMS) [2]. There is also a progressive form of the disease whereby disability accumulates independently of relapses, associated predominantly with neurodegeneration, named progressive MS (PMS) [2]. PMS is divided into primary PMS (PMS), which is infrequent, or secondary PMS (SPSM), which occurs as a late stage of RRMS occurring after years of disease inflammatory activity.
MS disease pathogenesis is complex, involving genetic susceptibility and environmental factors leading to the development of a pathologic autoimmune response against myelin, leading to myelin destruction, axonal loss, and focal inflammatory infiltrates [3]. MS appears to be the result of a misdirected immune response against one or several myelin proteins; however despite decades of research, no clear antigenic target has yet been identified as the cause [4,5].
Strong epidemiological data, particularly the high infection prevalence in patients with MS, suggests that Epstein-Barr virus (EBV) might be a prerequisite for MS development [6]. However, the underlying pathogenic mechanisms are still unclear [7]. EBV is a member of the Human Herpes Virus family that is transmitted via saliva and infects pharyngeal epithelial cells and B cells in the underlying tissue. In B-cells, EBV uncoats in the cytoplasm and transfers its DNA to the nucleus, where it can remain in a state of latency in which the viral genome can still be replicated along with cellular DNA. This state is called “deep” latency, given that the virus can be reactivated following B cell activation [8]. The mechanism by which EBV infection could trigger MS remains controversial [9]. EBV shares some structural properties with host proteins, so it might trigger self-perpetuating autoimmunity through molecular mimicry [10,11]. Alternatively, lack of control of persistent EBV infection might favor the establishment of a dysregulated immune response [9,12]. In any case, B lymphocytes may generate antibodies against EBV that, through molecular mimicry, may also attack the myelin and this could initiate an "active" phase of the disease.
Extracellular vesicles (EVs) are small bilayer membrane-wrapped particles that derive from the multivesicular body of their EV-originating cells and contain as cargo a wide variety of molecules. Recent research focuses on EVs as possible immune-mediators, given that they can cross the blood-brain barrier due to their small size, between 30 and 200nm, and carry a cargo implicated in disease pathogenesis [13]. Since EVs are readily available in the blood, they could serve as biomarkers in a wide variety of diseases [13,14], because they participate in the immune response and mimic the functions of their cells of origin. Due to their nature, EVs might be able to carry antibodies; however, this hypothesis has been scarcely explored. Recently, our group found that EVs of MS patients distinctively carry antibodies against myelin [15]. Antibodies are produced and secreted by B cells, and B-cells are known to secrete EVs, thus EVs could be mediating antibody transfer and may be detectable in the blood, constituting a potentially novel blood biomarker in MS [15].
We aimed to investigate whether antibodies against EBV were present in EVs of patients with MS and their possible usefulness as blood biomarkers. We measured levels of EV antibodies against EBV in terms of activity and neurological status in patients with RRMS to shed light on their possible role in disease pathogenesis. We also cross-examined EV antibody load for autoantibodies against myelin.”
Minor issues
Ttypos Sclerosis should not be capitalized (line 31). superscripts missing line 274.
Thank you for pointing this out. This has now been modified.
What is a detection threshhold of 3 mean ( line 276)
This is the machine detection threshold. If this threshold is set to 3, the machine analyzes circulating particles of 50nm and larger. With a lower threshold, it is able to detect smaller particles, but also more artifacts. In order to clarify this point, it now reads as follows (line 410):
“For the analysis, we employed a detection threshold of 3 in order to analyze 50nm vesicles or larger.”
This procedure was run in triplicate line 290 - clarify, were the same sample run on three lanes, or were three different samples analyzed.
We apologize for the misunderstanding. The same sample was run three times. Now it reads as follows (line 411):
“Each sample was run three times”
Methods 4.5 not much detail, seems like the same buffer as previous section.
We thank the reviewer for the comment, now the text reads as follows (lines 431-433):
“The EVs membrane was disrupted using a pierce radio-immunoprecipitation assay buffer (RIPA) (Thermo Scientific, USA) 1x dilution and 20 minutes incubation. To promote the EVs membrane rupture, vortexing was carried out every 5 minutes.”
The healthy controls provided informed, signed consent (line 250). It was not mentioned in methods that the patients provided informed, signed consent, although the statement at the end says everyone did. Should specifiy in methods.
The authors thank the reviewer for pointing this out. This is now specified at the end of the first paragraph of section 4.1 regarding patient inclusion and exclusion (lines 363-364).
Figure 1 A There is a white line cutting through image. B A loading control blot or stain should be shown to verify the plasma had sample on the membrane. More of the blot should be shown, in the figure or in supplemental to assess the possible crossreactivity of antibody, and to see the molecular weight markers. C) not clear why two graphs are shown, what are the colours (red, yellow, black, grey), D) missing scale bar.
Our apologies to the reviewers, we have deleted the line. B) We thank you for your contribution, now we show the original images with the controls and molecular weights in a supplementary figure. C) the graph on the left shows 3 lines: one orange, one red and one yellow, which represent each replicate measurement. The graph on the right represents the mean (in black) and SD (in red) of these above three. D) There is no scale because it is a screenshot of what the program analyzes in a video to determine the size of particles.
Text is now as follows:
Figure 1. EV Characterization. A) Electron microscopy showing EVs of <200nm. B) CD9, CD81, CD63 and Alix markers present in EVs. C) Graph showing EV sizes ranging 100-200nm. The sample was loaded three times shown in left panel (red, yellow and orange lines). The mean (black line) and SD (red line), are shown in the right panel. D) EVs as seen by NTA.
Discussion, line 175 claims that the results indicate EV antibodies can be diagnostic biomarker of MS, however, it is powered to be a pilot study, and the data shows considerable overlap between the MS and HC despite being significantly different, and there were few differences between active or stable.
We appreciate the remark and have now reworded this sentence to a more cautious form: “Our results indicate…” has been rephrased and now reads “Our results suggest..”
Also, a remark to underline the fact that this is a pilot study and results should be further validated in future cohorts has been added. Therefore, this now reads (lines 254-256):
“Our results suggest that antibodies against EVB found in EVs of patients could serve as a diagnostic biomarker in MS, however this is a pilot study and more studies should be carried out along these lines to analyze this.”
Comments on the Quality of English Language
Writing was very good. Just typo Figure 3 caption, VEB... perhaps meant to say EBV antibodies. Typo on x axis volumen
We thank the reviewer for the comment and for pointing out typos. They have now been corrected.
Reviewer 2 Report
Multiple sclerosis is a severe inflammatory demyelinating disease that dramatically affects the quality of life of patients. However, the etiology and pathogenesis of multiple sclerosis are still not completely clear. Epidemiological studies indicate that the Epstein-Barr virus may be a factor in the onset and development of this disease. Your article “Antibody content against Epstein-Barr virus in blood extracellular vesicles correlates with disease activity and brain volume in patients with relapsing remitting multiple sclerosis” is interesting and valuable. You have well described the differences between the concentration of antibodies in extracellular vesicles and in serum, indicating the possible influence of extracellular vesicles on the pathogenesis of the disease. It was also quite interesting to read about the correlations between the concentration of antibodies and the volume of various areas of the brain. I believe that such a manuscript is worthy of publication in the IJMS journal, however, I have a few comments and a few more minor ones.
Major suggestions
1. Please expand the "Introduction" section. The introduction section is very short. In my opinion, the introduction lacks information. Here are some examples.
1) Expand the paragraph about extracellular vesicles. How are they formed? How do antibodies appear in extracellular vesicles? How do they cross the BBB?
2) Expand the paragraph about RRMS. What other forms of multiple sclerosis are there and what is their relationship with the Epstein-Barr virus (if there is interesting information on this topic)? Why did you choose patients with this form of MS?
It is not necessary to write a lot. My examples are optional. But I think this section needs more useful information. You can decide to add whatever you want, I just gave an example of what I would like to see in the article.
2. Please write in detail the inclusion and exclusion criteria for MS patients in paragraph "4.1 Patient selection and evaluation" in the "Materials and Methods" section.
3. In paragraph “4.2 Sample collection and analysis” in the “Materials and Methods” section, you write only about plasma isolation. However, the manuscript states that you have conducted studies on both plasma and serum. Please write about serum isolation in the Materials and Methods section. Or, if you isolated only plasma, explain the appearance of serum in the study.
Minor suggestions
Line 31: Enter the acronym MS after the first mention of the disease in the text.
Line 46: Remove (Houen 2020), you have the link [7]. Or enter the author's last name in the sentence correctly.
Line 67: Please spell out the RRMS acronym when it first occurs. IJMS is a journal of general molecular profile and not all of its readers are familiar with neurological terminology.
Line 77: You write that the healthy controls were younger than the patients. Please write down their age as you did for patients (mean±SD).
Line 88: Table 1. Standard deviation is abbreviated as SD, not s. d. Replace please.
Line 88: Table 1. What does (seg) mean? This word occurs twice only in this table. Please write the designation in the description of the table.
Line 90: s. d. -> SD, as described above.
Line 90: In the description of the table, you have the standard deviation marked with the number 1 in upper case, but this designation is not found in the table.
Line 97: Spell out the acronym NTA, as it occurs here for the first time. This acronym is spelt out in the line 270 in the “Materials and Methods” section, where you can remove it and leave either the acronym or the full form.
Line 107; Line 108; Line 112: In the text, you have written the results in the form (mean ± SD), while the graphs are presented in the form of box plots, which show medians and quartiles. Please write the results as median [Q1; Q3] because the text does not reflect the figures.
Line 114: As far as I can understand, this is not about healthy controls, but about stable patients, since this paragraph is about comparing different groups of patients. Please replace healthy controls with stable patients (or whatever you want to call them). I hope I understood this part of the manuscript correctly and did not mislead you.
Line 129: In the description of Figure 3, you use the acronym VEB. Perhaps here you mean EBV?
Line 132; Line 136; Line 137; Line 138; Line 139: Speaking of correlation, you write R-0.4 everywhere; R-0.3; R-0.3 and so on. Because of this spelling, it is visually impossible to distinguish a positive correlation from a negative one. It would be more correct to write, say, R = 0.4, if we are talking about a positive correlation, and R = -0.3, if we are talking about a negative correlation.
Line 139: (R R-0.3;p=0.03) remove the extra R.
Line 155 — Line 158: The first three sentences in this paragraph are essentially a repetition of what you already wrote in the "Introduction" section. Remove it or rewrite it so that it is not so noticeable.
Line 249; Line 250. You write that your healthy controls were blood donors. I think it's worth pointing out in line 218 that the healthy controls were not only younger, but were also blood donors. This fact may also slightly limit the study, since various blood parameters, including immunological ones, may change in regular blood donors. I don't see this as a major limitation, but I think it's worth mentioning.
Line 255; Line 256; Line 280; Line 281: Try not to use the word "We", rewrite these sentences without this word.
Throughout the manuscript: Somewhere you write EBNA-1, and somewhere EBNA1, please use some uniform format.
Good luck with your future work.
Best regards
Author Response
Reviewer 2
Multiple sclerosis is a severe inflammatory demyelinating disease that dramatically affects the quality of life of patients. However, the etiology and pathogenesis of multiple sclerosis are still not completely clear. Epidemiological studies indicate that the Epstein-Barr virus may be a factor in the onset and development of this disease. Your article “Antibody content against Epstein-Barr virus in blood extracellular vesicles correlates with disease activity and brain volume in patients with relapsing remitting multiple sclerosis” is interesting and valuable. You have well described the differences between the concentration of antibodies in extracellular vesicles and in serum, indicating the possible influence of extracellular vesicles on the pathogenesis of the disease. It was also quite interesting to read about the correlations between the concentration of antibodies and the volume of various areas of the brain. I believe that such a manuscript is worthy of publication in the IJMS journal, however, I have a few comments and a few more minor ones.
The authors thank reviewer 2 for underlining the interest and value of our work. We are also very thankful for his/her time and helpful comments that have greatly contributed to ameliorate the quality of the manuscript.
Major suggestions
- Please expand the "Introduction" section. The introduction section is very short. In my opinion, the introduction lacks information. Here are some examples.
1) Expand the paragraph about extracellular vesicles. How are they formed? How do antibodies appear in extracellular vesicles? How do they cross the BBB?
2) Expand the paragraph about RRMS. What other forms of multiple sclerosis are there and what is their relationship with the Epstein-Barr virus (if there is interesting information on this topic)?Why did you choose patients with this form of MS?
It is not necessary to write a lot. My examples are optional. But I think this section needs more useful information. You can decide to add whatever you want, I just gave an example of what I would like to see in the article.
The authors want to thank reviewer 2 for his/her suggestions on the introduction, as well as for pointing out the need for a better explanation of MS and its disease forms, taking into account not all of its readers are familiar with neurological terminology. We have explained that initially, at disease onset, RRMS is the most frequent type of MS as PPMS is rare and SPMS is secondary to RRMS. Also, EBV is considered a disease trigger. EBV virus is a virus that remains latent in B lymphocytes and, when activated, it is suggested that B lymphocytes may generate antibodies against this virus that, through molecular mimicry, may also attack the myelin and this could initiate an "active" phase of the disease. That is why we have chosen RRMS, to be able to study the active phase vs. the non-active phase. As this hypothesis was not understood in the previous version of the manuscript, we have also added a new experiment analyzing antibodies against myelin proteins in EVs and studying possible correlations with EV content of EBV antibodies in the new version of the manuscript to help the readers shed light on the hypothesis.
Most literature revieweing evidence on EBV and MS, talks about MS as a single entity, as EBV infection has been linked to both RRMS and PMS, albeit with many different theories. This is why are results are also of interest analyzing EV content in antibodies against EBV exploring disease activity, and also time from disease onset.
Also, more information is provided on extracellular vesicles regarding vesicle formation, antibody content and vesicle size (allowing the to cross the blood-brain barrier).
The introduction now reads as follows:
“Multiple sclerosis (MS) is an inflammatory demyelinating disease of the central nervous system (CNS) with multifocal areas of demyelination [1]. There are two main different disease forms, with the most common being a relapsing disease where periods of inflammatory activity alternate with periods of disease inactivity, named relapsing-remitting MS (RRMS) [2]. There is also a progressive form of the disease whereby disability accumulates independently of relapses, associated predominantly with neurodegeneration, named progressive MS (PMS) [2]. PMS is divided into primary PMS (PMS), which is infrequent, or secondary PMS (SPSM), which occurs as a late stage of RRMS occurring after years of disease inflammatory activity.
MS disease pathogenesis is complex, involving genetic susceptibility and environmental factors leading to the development of a pathologic autoimmune response against myelin, leading to myelin destruction, axonal loss, and focal inflammatory infiltrates [3]. MS appears to be the result of a misdirected immune response against one or several myelin proteins; however despite decades of research, no clear antigenic target has yet been identified as the cause [4,5].
Strong epidemiological data, particularly the high infection prevalence in patients with MS, suggests that Epstein-Barr virus (EBV) might be a prerequisite for MS development [6]. However, the underlying pathogenic mechanisms are still unclear [7]. EBV is a member of the Human Herpes Virus family that is transmitted via saliva and infects pharyngeal epithelial cells and B cells in the underlying tissue. In B-cells, EBV uncoats in the cytoplasm and transfers its DNA to the nucleus, where it can remain in a state of latency in which the viral genome can still be replicated along with cellular DNA. This state is called “deep” latency, given that the virus can be reactivated following B cell activation [8]. The mechanism by which EBV infection could trigger MS remains controversial [9]. EBV shares some structural properties with host proteins, so it might trigger self-perpetuating autoimmunity through molecular mimicry [10,11]. Alternatively, lack of control of persistent EBV infection might favor the establishment of a dysregulated immune response [9,12]. In any case, B lymphocytes may generate antibodies against EBV that, through molecular mimicry, may also attack the myelin and this could initiate an "active" phase of the disease.
Extracellular vesicles (EVs) are small bilayer membrane-wrapped particles that derive from the multivesicular body of their EV-originating cells and contain as cargo a wide variety of molecules. Recent research focuses on EVs as possible immune-mediators, given that they can cross the blood-brain barrier due to their small size, between 30 and 200nm, and carry a cargo implicated in disease pathogenesis [13]. Since EVs are readily available in the blood, they could serve as biomarkers in a wide variety of diseases [13,14], because they participate in the immune response and mimic the functions of their cells of origin. Due to their nature, EVs might be able to carry antibodies; however, this hypothesis has been scarcely explored. Recently, our group found that EVs of MS patients distinctively carry antibodies against myelin [15]. Antibodies are produced and secreted by B cells, and B-cells are known to secrete EVs, thus EVs could be mediating antibody transfer and may be detectable in the blood, constituting a potentially novel blood biomarker in MS [15].
We aimed to investigate whether antibodies against EBV were present in EVs of patients with MS and their possible usefulness as blood biomarkers. We measured levels of EV antibodies against EBV in terms of activity and neurological status in patients with RRMS to shed light on their possible role in disease pathogenesis. We also cross-examined EV antibody load for autoantibodies against myelin.”
Of note, the last sentence of the introduction section now reads “We also cross-examined EV antibody load for autoantibodies against myelin” because, to further analyze possible differences in antibody content of EVs between MS patients and healthy controls, we have performed a new experiment analyzing antibodies against myelin proteins in EVs and studying possible correlations with EV content of EBV antibodies.
Results are provided in sections 2.8 (and figure 4 which has now been included in the manuscript):
“2.8. Antibodies against myelin basic protein (MBP) in EVs of MS patients correlated with EV anti-VCA content.
Anti-MBP content of EVs correlated with EV Anti-VCA content (R=0.36; p=0.006). No correlation was found in EV anti-MBP content with anti-VCA titters in plasma or with anti-EBNA1 titters in EVs or plasma. Also, no correlation was found in anti-MOG content of EVs or plasma with antibodies against EBV or in EV content of a control protein (albumin) and EV antibodies against EBV.”
And these results are commented in the discussion section (lines 256-263):
“Moreover, in the present study we observe a correlation between anti-VCA and an-ti-MBP content in EVs, but not with other myelin proteins (MOG) or control proteins (al-bumin). Given that EBV shares some structural properties with myelin proteins, this result may indicate that EVs could be loading both, anti-EBV and anti-MBP antibodies, to attack myelin during disease. On the other hand, recently antibodies against myelin have been found in EVs of MS patients in significantly higher levels than in healthy controls [15]. This argues in favor of the possible usefulness of anti-VCA content of EVs as a blood bi-omarker in MS, suggesting that the EVs that load antibodies against EBV may also, through cross reactivity, load autoantibodies against myelin antigens.”
2.Please write in detail the inclusion and exclusion criteria for MS patients in paragraph "4.1 Patient selection and evaluation" in the "Materials and Methods" section.
The authors greatly thank the reviewer for pointing out that this paragraph was missing. We have now included this information in section 4.1 as indicated (lines..).
“Patients with RRMS were recruited sequentially, including both patients with active disease (defined by an ongoing relapse requiring steroid therapy) and patients without disease activity (NEDA 3 criteria: clinically and radiological stability for over a year). For comparison, we also included healthy controls. Patient inclusion criteria were: diagnosis of RRMS, according to McDonald criteria [29]. and age of ≥18 years. Patients were included before the start of corticosteroid treatment, steroid treatment in the prior 2 months was an exclusion criteria. Other exclusion criteria were progressive disease, pregnancy, lactation, drug or alcohol dependence, severe concomitant disease or another autoimmune disease, and participation in a clinical trial. All patients signed informed consent on inclusion.”
- In paragraph “4.2 Sample collection and analysis” in the “Materials and Methods” section, you write only about plasma isolation. However, the manuscript states that you have conducted studies on both plasma and serum. Please write about serum isolation in the Materials and Methods section. Or, if you isolated only plasma, explain the appearance of serum in the study.
We apologize for the error, only plasma was used. This has now been changed throughout the manuscript.
Minor suggestions
Line 31: Enter the acronym MS after the first mention of the disease in the text.
The authors thank the reviewer for realising this was missing. It has now been included (line 31).
Line 46: Remove (Houen 2020), you have the link [7]. Or enter the author's last name in the sentence correctly.
Thak you for realising this was redundant. It has now been deleted (line 54).
Line 67: Please spell out the RRMS acronym when it first occurs. IJMS is a journal of general molecular profile and not all of its readers are familiar with neurological terminology.
Thank you for highlighting the journal’s reader profile. As mentioned we have now expanded the introduction section following your suggestions. Disease types are explained, including RRMS, which has been spelled out followed by the acronym in the first paragraph of the article.
Line 77: You write that the healthy controls were younger than the patients. Please write down their age as you did for patients (mean±SD).
Thank you for the comment, this has been included:
“There was no difference in age between stable patients (43.4 ±7.7 years) and patients in relapse (40.2 ± 8.4 years); however healthy controls were younger than patients with MS (29.9 ± 8.7).”
Line 88: Table 1. Standard deviation is abbreviated as SD, not s. d. Replace please.
The authors thank the reviewer for pointing this out. Abbreviation has been replaced appropriately.
Line 88: Table 1. What does (seg) mean? This word occurs twice only in this table. Please write the designation in the description of the table.
The authors thank the reviewer for pointing out this typo and the missing description. It has been corrected, and the description added in the legend.
Line 90: s. d. -> SD, as described above.
The authors thank the reviewer for pointing this out. Abbreviation has been replaced appropriately.
Line 90: In the description of the table, you have the standard deviation marked with the number 1 in upper case, but this designation is not found in the table.
Thank you for pointing this out. We have deleted number 1 as it was a typing error.
Line 97: Spell out the acronym NTA, as it occurs here for the first time. This acronym is spelt out in the line 270 in the “Materials and Methods” section, where you can remove it and leave either the acronym or the full form.
Thank you for pointing out that NTA should be spelled out in the results setion for comprehension as it comes first in the paper.
Line 107; Line 108; Line 112: In the text, you have written the results in the form (mean ± SD), while the graphs are presented in the form of box plots, which show medians and quartiles.Please write the results as median [Q1; Q3] because the text does not reflect the figures.
We have now adapted the text as suggested to reflect the figures.
It now reads as follows:
“2.3. Higher EV anti-VCA titers were found in patients compared with healthy controls
We found that patients with RRMS had higher titers of anti-VCA, both inside EVs and free in plasma, than healthy controls (120.7 [Q1=84.0; Q3=166.0] UR/mL vs. 59.1 [Q1=33.0; Q3=125.8] UR/mL, p=0.001; 121.9 [Q1=91.5; Q3=152.2] UR/mL vs. 74.7 [Q1=58.8; Q3=97.3] UR/mL, p≤0.001). We found no differences in anti-EBNA1 titers in plasma or EVs between patients with MS or healthy controls (Figure 2A).
2.4 Higher EV anti-EBNA1 antibodies levels were found in patients with disease activity
When comparing patients according to disease activity, we found that patients with active disease had higher titers of anti-EBNA1 in EVs (110.4 [Q1=69.6; Q3=146.9] UR/mL vs. 84.3 [Q1=18.5; Q3=111.4] UR/mL; p=0.027), but not in plasma. We did not find significant differences in anti-VCA content in patients with active disease compared to patients without disease activity, albeit the former had a tendency toward higher anti-VCA levels. (Figure 2B).”
Line 114: As far as I can understand, this is not about healthy controls, but about stable patients, since this paragraph is about comparing different groups of patients. Please replace healthy controls with stable patients (or whatever you want to call them). I hope I understood this part of the manuscript correctly and did not mislead you.
The authors greatly thank the reviewer for noticing this error. We have now replaced healthy controls with patients without disease activity.
Line 129: In the description of Figure 3, you use the acronym VEB. Perhaps here you mean EBV?
Thank you for pointing out this typing error. It has now been corrected.
Line 132; Line 136; Line 137; Line 138; Line 139: Speaking of correlation, you write R-0.4 everywhere; R-0.3; R-0.3 and so on. Because of this spelling, it is visually impossible to distinguish a positive correlation from a negative one. It would be more correct to write, say, R = 0.4, if we are talking about a positive correlation, and R = -0.3, if we are talking about a negative correlation.
The authors thank the reviewer for pointing this out. We have rewritten correlations as suggested.
Line 139: (R R-0.3;p=0.03) remove the extra R.
Thank you for pointing out typing errors. The extra R has now been removed.
Line 155 — Line 158: The first three sentences in this paragraph are essentially a repetition of what you already wrote in the "Introduction" section. Remove it or rewrite it so that it is not so noticeable.
The authors have rewritten these sentences, summarizing the information provided. We have not removed it because we believe it helps with manuscript comprehension if a reader were to skip the introduction.
Line 249; Line 250. You write that your healthy controls were blood donors. I think it's worth pointing out in line 218 that the healthy controls were not only younger, but were also blood donors. This fact may also slightly limit the study, since various blood parameters, including immunological ones, may change in regular blood donors. I don't see this as a major limitation, but I think it's worth mentioning.
The authors thank the reviewer for highlighting the interest of commenting on this point.The following sentence has been included (lines 238-247):
“Regarding healthy controls, it must also be noted that this study employed healthy blood donors as controls. It has been postulated that regular whole blood donation may have an impact on individual blood donors, particularly regarding cell counts and iron content [34]. However, recent mass spectrometry based studies have shown that only sparse changes occur in plasma proteomes of new and regular donors after a whole blood donation [35].”
- Furuta M, Shimizu T, Mizuno S et al. Clinical evaluation of repeat apheresis donors in Japan. Vox Sang. 1999;77(1):17-23.
- Kreft IC, Hoogendijk AJ, van der Zwaan C et al. Mass spectrometry-based analysis on the impact of whole blood donation on the global plasma proteome. Transfusion. 2023 Mar;63(3):564-573.
Line 255; Line 256; Line 280; Line 281: Try not to use the word "We", rewrite these sentences without this word.
These lines have been rewritten as suggested.
Throughout the manuscript: Somewhere you write EBNA-1, and somewhere EBNA1, please use some uniform format.
Thank you for pointing this out. We have now verified that only EBNA1 is written throughout the manuscript.
Good luck with your future work.
Best regards
The authors are very thankful not only for the work and comments of the reviewer but also for the reviewer’s wishes. We are enthusiastic about the window that this pilot study opens and will continue our research in this area.

Round 2
Reviewer 1 Report
In the revised article, the authors clarified key clinical definitions with explanations and several new references, and edited the introduction and discussion points accordingly. It may require a longer assessment period to determine disease stability. As they pointed out, the EDSS scores were similar between the groups and the main differences were the activity versus stability using 1 year and disease duration. With the information now presented, readers can understand and draw their own conclusions from the interesting results on EV antibody diagnostics. Other editorial issues were fixed.